# Comparative Study of Cardiovascular Effects of Selected Pulmonary Vasodilators in Canine Models of Mitral Valve Disease

**DOI:** 10.3390/biology13050311

**Published:** 2024-04-30

**Authors:** Yunosuke Yuchi, Ryohei Suzuki, Narumi Ishida, Shuji Satomi, Takahiro Saito, Takahiro Teshima, Hirotaka Matsumoto

**Affiliations:** 1Laboratory of Veterinary Internal Medicine, School of Veterinary Medicine, Faculty of Veterinary Science, Nippon Veterinary and Life Science University, Tokyo 180-8602, Japan; y.0301.yunosuke@gmail.com (Y.Y.); jetlog21117@yahoo.co.jp (S.S.); justthe2ofussaito@yahoo.co.jp (T.S.); teshima63@nvlu.ac.jp (T.T.); matsumoto@nvlu.ac.jp (H.M.); 2Garden Veterinary Hospital, Tokyo 153-0063, Japan

**Keywords:** beraprost sodium, cardiac output, dog, mitral valve regurgitation, pulmonary hypertension, pulmonary arterial pressure, right-heart catheterization, sildenafil, speckle tracking echocardiography, vascular resistance

## Abstract

**Simple Summary:**

Pulmonary hypertension is a fatal comorbidity in dogs with left-sided heart disease. Various oral pulmonary vasodilators have been effective in treating canine pulmonary hypertension; however, no studies have compared their hemodynamic effects. This study compared the hemodynamic effects of selected pulmonary vasodilators (15 µg/kg beraprost, 1.0 mg/kg sildenafil, and their combination) in canine models of mitral regurgitation (the most common cardiac disease in dogs). Significant improvements in pulmonary hypertension were observed with all study drugs. Pulmonary vasodilating effects differed among the study’s drugs. Sildenafil showed a more potent pulmonary vasodilating effect than beraprost; however, sildenafil significantly worsened the left-heart loading condition. Although beraprost showed a weaker pulmonary vasodilating effect than sildenafil, no significant worsening in the left-heart loading condition was observed. Combination therapy resulted in the strongest pulmonary and systemic vasodilating effects without worsening the left-heart loading condition. This study demonstrated the differences between beraprost and sildenafil in pulmonary and systemic vasodilating effects. Sildenafil involved the risk of worsening the left-heart load, although it was effective in treating pulmonary hypertension. Combination therapy with beraprost and sildenafil synergistically dilated the pulmonary and systemic vessels, indicating a more potent treatment option.

**Abstract:**

Previous reports have shown that various oral pulmonary vasodilators are effective against canine pulmonary hypertension (PH). However, no studies have compared their hemodynamic effects. We aimed to compare the hemodynamic effects of 15 µg/kg beraprost sodium, 1.0 mg/kg sildenafil, and their combination, in dogs with experimentally induced mitral regurgitation. This experimental crossover study evaluated the hemodynamic and functional effects of oral pulmonary vasodilators by application of right-sided heart catheterization and echocardiography. Beraprost significantly decreased pulmonary and systemic vascular resistance. Additionally, beraprost increased right-ventricular stroke volume and left-ventricular cardiac output without worsening left-heart size and left-atrial pressure. The pulmonary vasodilatory effects of sildenafil were stronger, and its systemic vasodilatory effects were weaker than those of beraprost. However, sildenafil significantly increased the left-ventricular volume, left-atrial pressure indicator, and right-ventricular cardiac output. Combination therapy resulted in the strongest pulmonary and systemic vasodilating effects without worsening the left-heart size and left-atrial pressure indicators. Both beraprost and sildenafil were effective against canine PH; however, sildenafil was associated with the risk of worsening left-heart loading. Combination therapy with beraprost and sildenafil synergistically dilated pulmonary and systemic vessels, indicating a more potent treatment option for severe PH cases.

## 1. Introduction

Pulmonary hypertension (PH) is a life-threatening condition in patients with cardiopulmonary diseases [1,2]. This disease can cause pressure overload on the right side of the heart, resulting in right-ventricular (RV) hypertrophy, dilatation, and dysfunction [2,3,4,5]. In humans, left-sided heart disease is the most common cause of PH next to pulmonary venous congestion [6]. Additionally, myxomatous mitral valve disease is the most common cause of PH in dogs [7,8]. Concurrent PH complicates pulmonary and systemic hemodynamics and can confound treatment. According to the guidelines for managing PH, the primary recommended treatment for PH due to left-heart disease is to treat the underlying disease (i.e., decrease left-atrial pressure and, if present, left-heart failure) [1,2]. However, pulmonary vasodilators are typically considered in cases of PH that cannot be controlled solely by treating underlying diseases in dogs [1].

Previous veterinary medicine reports showed that several oral pulmonary vasodilators were effective for PH, including phosphodiesterase 5 inhibitors (sildenafil and tadalafil) [9,10,11], endothelin receptor blockers (ambrisentan) [12,13], and prostaglandin I_2_ inhibitors (beraprost sodium (BPS)) [14,15,16]. Although sildenafil is the pulmonary vasodilator most frequently reported as effective for PH in dogs, some studies showed that the left-heart loading condition could worsen due to excessive pulmonary circulation [1,17,18,19]. The hemodynamic changes caused by sildenafil are a serious concern in PH treatment, especially in dogs with PH secondary to left-sided heart disease. Although various pulmonary vasodilators cause vasodilatory effects through different mechanisms, no studies have compared the hemodynamic effects of different oral pulmonary vasodilators in dogs. On the other hand, combination therapy using pulmonary vasodilators with different mechanisms is widely used in human patients with PH [2,20].

This study compared the hemodynamic and functional effects of selected pulmonary vasodilators (BPS and sildenafil), and evaluated the clinical efficacy of combination therapy with BPS and sildenafil in dogs with experimentally induced mitral regurgitation (MR). We hypothesized that the degree of pulmonary vasodilating effects of BPS and sildenafil were different, and that sildenafil could improve PH pathophysiology but worsen left-heart loading conditions. Furthermore, combination therapy with BPS and sildenafil could be a more potent treatment for PH, owing to their synergistic effects.

## 2. Materials and Methods

This was a hypothesis-driven prospective crossover study conducted in accordance with the Guide for Institutional Laboratory Animal Care and Use at Nippon Veterinary and Life Science University, Japan. All study procedures were approved by the Ethics Committee for Laboratory Animal Use at Nippon Veterinary and Life Science University, Japan (approval number: 2022S-10).

### 2.1. Animals and Study Preparation

Six laboratory-owned male beagles were enrolled in this study (age: 2.4 ± 0.1 years; body weight: 11.9 ± 0.8 kg). In 3 of the 6 dogs, a left-lateral thoracotomy was performed in the fifth intercostal space, and MR was experimentally induced by rupturing the chordae tendineae attached to the mitral valve by inserting mosquito forceps from the incised left-atrial appendage (*n* = 3) [21]. In the other three dogs, MR was induced by rupturing the chordae tendineae by inserting myocardial biopsy forceps into the carotid artery (*n* = 3). In both methods, the degree of MR was monitored using transthoracic two-dimensional and color-Doppler echocardiography, and rupture of the chordae tendineae was repeated until a distinct MR was observed (regurgitation jet signals occupying approximately 30% of the left atrium were observed on color-Doppler echocardiography) [21].

### 2.2. Study Protocol

Before administration of the study drugs (BPS, sildenafil, or BPS + Sildenafil [BPS + Sil]), all 6 dogs underwent right-heart catheterization and transthoracic echocardiography. To perform these procedures, dogs were administered butorphanol tartrate (Meiji Animal Health Co., Ltd., Tokyo, Japan; 0.1 mg/kg, IM), midazolam hydrochloride (Maruishi Pharmaceutical. Co., Ltd., Osaka, Japan; 0.1 mg/kg, IM), and ampicillin sodium (VMDP Co., Ltd., Tokyo, Japan; 20 mg/kg, IV). The dogs were then restrained in left-lateral recumbency, and the right side of the neck around the right jugular vein area was clipped, aseptically prepared, and draped. Local anesthesia was administered around the right jugular vein using lidocaine (Shionogi Pharma Co., Ltd., Osaka, Japan), and a 6 Fr sheath introducer (Radifocus Introducer IIH, Terumo Corporation, Tokyo, Japan) was inserted into the right jugular vein using the Seldinger method. Hemodynamic measurements were performed using a 4 Fr thermodilution catheter (Edwards Lifesciences Corporation, Tokyo, Japan). After completing right-heart catheterization, the thermodilution catheter and sheath introducer were removed, and astriction was performed manually. Echocardiography was immediately performed.

After performing the above examinations, dogs were continuously administered one of the following oral drugs twice a day for 1 week: 15 µg/kg BPS (using 55 µg tablets), 1.0 mg/kg sildenafil (using 20 mg tablets), or the combination of drugs. Doses of BPS and sildenafil were determined according to a previous report [14,15,22]. Each drug was tested using all 6 dogs by a crossover method. The same examinations were performed after continuous administration. A 1-week washout period was provided after completing post-examinations of each patient and drug. Upon completion of the study protocol, all dogs were transferred to another protocol.

### 2.3. Hemodynamic Measurement

Hemodynamic measurements were performed by a single investigator (Y.Y.) using a thermodilution catheter (Edwards Lifesciences Corporation, Tokyo, Japan) and hemodynamic analysis software (LabChart Pro version 7.3.8; ADInstruments, Nagoya, Aichi, Japan). All hemodynamic analyses were performed by another investigator (N.I.). Lead II electrocardiography was also displayed simultaneously throughout the hemodynamic measurements. Additionally, the systolic and mean systemic arterial pressure (SAP and MAP, respectively; mmHg) were measured using an oscillometric method (BP100D; Fukuda M-E Kogyo Co., Ltd., Chiba, Japan).

The following hemodynamic variables were measured in the same order: pulmonary capillary wedge pressure (PCWP; mmHg); pulmonary arterial pressure at systole, mean, and diastole (sPAP, mPAP, and dPAP, respectively; mmHg); RV pressure at systole and diastole (sRVP and dRVP, respectively; mmHg); mean right-atrial pressure (RAP; mmHg); and mean central venous pressure (CVP; mmHg). Using the RV pressure wave, the maximal and minimal first derivatives of RVP (dP/dt_max_ and dP/dt_min_, respectively; mmHg/s) were calculated. All variables were obtained from 10 consecutive sinus rhythms, and the average values were used for statistical analyses. Cardiac output (CO; L/min) was then obtained using a thermodilution technique, in which 5 mL of ice-cold saline solution was injected rapidly from the proximal port. Additionally, stroke volume (SV; mL) was calculated by dividing the CO by the heart rate at the same time. The average value of three measurements was used for statistical analyses. Pulmonary and systemic vascular resistance (PVR and SVR, respectively) were calculated using the following formulas:PVR = (mPAP − PCWP)/RV CO(1)
SVR = (mSAP − RAP)/Left ventricular CO(2)

In this formula, left-ventricular (LV) CO was measured by echocardiography using the cross-sectional area method described below [23].

### 2.4. Echocardiography

Two-dimensional Doppler echocardiography was performed using an echocardiographic system (Vivid E95 Ultra Edition; GE Healthcare, Tokyo, Japan) equipped with a 3.5–6.9 MHz transducer by the investigator who performed the right-heart catheterization (Y.Y.). The investigator who performed the hemodynamic analyses (N.I.) carried out all echocardiographic analyses using an offline workstation (EchoPAC PC, Version 204; GE Healthcare, Tokyo, Japan). Mean values from three consecutive sinus rhythms were used for statistical analyses.

Immediately after right-heart catheterization, the sedated dogs were manually restrained in left- and right-lateral recumbency and underwent echocardiography. The following variables were obtained as indices of the left-heart morphology: left-atrial-to-aortic diameter ratio (LA/Ao) and body-surface-area-normalized LV volume at end-diastole and end-systole (LVEDV and LVESV, respectively; mL/m^2^). The LA/Ao ratio was measured using the right parasternal short-axis view at the level of the base of the heart [24]. The LV volume was obtained by the biplane area-length method using left-apical four-chamber and two-chamber views at similar heart rates [25]. Additionally, the LV ejection fraction (EF; %) was calculated using LVEDV and LVESV [25]. For left-heart hemodynamics, early-diastolic and late-diastolic transmitral flow velocities (m/s) were measured using the left-apical four-chamber view and pulsed-wave Doppler method (E and A, respectively). Tissue Doppler imaging-derived early-diastolic myocardial velocity of the septal mitral annulus (e’; cm/s) was measured. Using these hemodynamical variables, E/A and E/e’ were also calculated as the left-atrial pressure indicators [26]. Furthermore, LV CO (L/min) was calculated using the cross-sectional area method and the following variables: aortic valve diameter, velocity-time integral of the systemic arterial flow, and heart rate [23].

For right-heart morphology, the RV area at end-diastole and end-systole (RVEDA and RVESA, respectively; cm^2^) was obtained using the left-apical four-chamber view optimized for the right heart (RV focus view), as reported previously [27,28]. The RV function was evaluated using the following formulas: RV fractional area change (FAC; %), tricuspid annular plane systolic excursion (TAPSE; mm), and tissue Doppler imaging-derived systolic myocardial velocity of the lateral tricuspid annulus (RV s’; cm/s). TAPSE was measured using the B-mode method as described previously [29]. The RV FAC and TAPSE levels were normalized to body weight (RV FACn and TAPSEn, respectively) [29,30].
RV FACn = (RVEDA − RVESA)/RVEDA × 100/(body weight [kg])^−0.097^,(3)
TAPSEn = (TAPSE [mm])/(body weight [kg])^0.284^(4)

In this study, myocardial function was evaluated using two-dimensional speckle-tracking echocardiography as a detailed indicator of systolic function. LV function was evaluated using LV longitudinal and circumferential strains (LV-SL and LV-SC, respectively; %) [15,31]. LV-SL was obtained from the left-apical four-chamber view, which was used to measure LV volume and EF. LV-SC was obtained from the right parasternal short-axis view at the level of the papillary muscle. Additionally, as an RV myocardial function indicator, RV longitudinal strain (RV-SL; %) was measured using the RV focus view, which was used to measure other RV echocardiographic variables. RV-SL was measured using the following two patterns: RV-SL from only the RV free wall (RV-SL_3seg_) and RV-SL from the RV free wall and interventricular septum (RV-SL_6seg_) [14].

### 2.5. Statistical Analyses

Data are expressed as the median (interquartile range). All statistical analyses were performed using commercially available software (EZR version 1.62) [32]. The normality of the data was evaluated using the Shapiro–Wilk test. The effects of BPS, sildenafil, and BPS + Sil on hemodynamic and echocardiographic variables were validated by a paired *t*-test for normally distributed data or Wilcoxon signed-rank test for non-normally distributed data. A *p*-value < 0.05 was considered statistically significant.

## 3. Results

All procedures were completed for all dogs in the MR model. No dogs showed any clinical signs of congestive heart failure throughout the study period. The time from induction of MR to the beginning of the study ranged from 295 to 640 days. At the beginning of the study, all canine models had moderate to severe MR and left-heart enlargement without any evidence of heart failure, indicating American College of Veterinary Internal Medicine Stage B2 (mean ± standard deviation of LA/Ao and the normalized end-diastolic LV internal dimensions at the beginning of the study were 1.7 ± 0.1 and 1.8 ± 0.1, respectively).

The hemodynamic variables before and after the administration of each pulmonary vasodilator are summarized in Table 1 and Figure 1. Systemic arterial pressure, RAP, and CVP were unchanged by the administration of each drug. All examined drugs significantly decreased sPAP, mPAP, and PVR (BPS: *p* = 0.03, 0.046, and 0.04, respectively; sildenafil: *p* = 0.03, 0.046, and 0.01, respectively; BPS + Sil: *p* < 0.01, 0.04, and 0.03, respectively). Additionally, SVR also decreased after the administration of BPS and BPS + Sil (*p* = 0.046 and 0.01, respectively). A comparison of the decreasing rates of PVR and SVR showed that the pulmonary vasodilating effect of sildenafil was stronger than that of BPS, and demonstrated a pulmonary vasopredominant effect. In contrast, the BPS group showed a decreased PVR and SVR. The BPS + Sil group showed the strongest vasodilatory effect (Figure 2). BPS significantly decreased sRVP (*p* = 0.01). Although PCWP did not change after the administration of BPS or BPS + Sil, sildenafil significantly increased it (*p* = 0.045). BPS significantly increased RV SV (*p* = 0.01). Both sildenafil and BPS + Sil significantly increased dP/dt_max_ (both *p* = 0.04), whereas only BPS + Sil increased RV CO (*p* = 0.03).

Echocardiographic variables of the left heart before and after the administration of each pulmonary vasodilator are summarized in Table 2. No significant changes were observed in LA/Ao, A, E/A, and LV-SC. BPS and BPS + Sil significantly increased LV CO (*p* = 0.01 and 0.04, respectively). Additionally, sildenafil significantly increased LVEDV, LVESV, E/e’ (*p* < 0.01, 0.02, and 0.03, respectively), and LV-SL was also significantly increased (*p* < 0.01).

The right-heart hemodynamics and functions estimated by using echocardiography are summarized in Table 3. We found no significant changes in the RV area after the administration of sildenafil, except for the RVESA (*p* = 0.01). Regarding RV function, BPS showed no significant improvement in RV functional variables. Sildenafil significantly increased RV FACn (*p* < 0.01), and BPS + Sil significantly increased RV-SL (RV-SL_3seg_: *p* = 0.02; RV-SL_6seg_: *p* = 0.03).

## 4. Discussion

This is the first study (to our knowledge) to compare the hemodynamic effects of selected pulmonary vasodilators using canine models of mitral regurgitation (mitral valve disease equivalent to American College of Veterinary Internal Medicine Stage B2 [33]) and mildly elevated PAP. All drugs examined in this study significantly decreased invasively measured PAP. When comparing BPS and sildenafil, the findings showed that BPS decreased PVR and SVR equally, whereas 1 mg/kg sildenafil caused a stronger decrease in PVR and a weaker decrease in SVR, indicating a pulmonary vasopredominant effect. This difference suggests that sildenafil might worsen left-heart loading conditions. The combination of BPS and sildenafil further decreased the PVR through synergistic effects and reduced the excessive left-heart loading caused by sildenafil through the systemic vasodilating effect of BPS.

In this study, we observed differences in the vasodilating effect between 15 µg/kg BPS and 1.0 mg/kg sildenafil, although both BPS and sildenafil significantly decreased PVR and SVR. The 15 µg/kg BPS decreased PVR and SVR equally, whereas sildenafil showed a pulmonary vasopredominant effect and decreased PVR more strongly. Therefore, the results suggested that although both BPS and sildenafil would be effective for PH, the pulmonary vasodilating effect might be more potent in 1.0 mg/kg sildenafil than 15 µg/kg BPS. However, 1.0 mg/kg sildenafil significantly increased LV volume and E/e’, indicators of left-atrial pressure. These results are in line with those of previous reports [1,17,18,19] and suggest that careful administration of sildenafil is recommended in patients with PH secondary to left-heart disease. In contrast, although BPS also improved RV hemodynamics based on RV SV, significant worsening of the left-heart size and left-atrial pressure was not observed in the canine MR models. This difference may be attributable to the degree of systemic vasodilation. A previous report suggested that amlodipine, a systemic vasodilator, significantly decreased left-atrial pressure and increased cardiac output from the left ventricle [34]. In this study, BPS administration significantly decreased SVR and increased LV CO without worsening left-heart loading, possibly due to its systemic vasodilatory effect. These results indicated that BPS could potentially treat canine PH caused by left-heart disease more safely than sildenafil. However, BPS has reported animal species differences in the pharmacologic effects [35]. Our results might be different in species other than dogs, as it has been reported in the PH guidelines that BPS showed no hemodynamic improvements or long-term outcome benefits in human patients with PH [2].

The RV functional indicators in this study, except for RV SV, were not improved after treatment with 15 µg/kg BPS, although they showed a tendency toward improvement. In previous reports using BPS in chronic embolic PH dog models and clinical cases with various causes, BPS significantly improved RV functional indicators, such as TAPSE, RV FAC, and RV-SL [14,15]. The discrepancy between these results may be attributed to the severity of PH. In previous reports, dogs with moderate PH had worse RV functional indicators. In contrast, the dogs in this study had mild PH based on invasive PAP measurements. Because the right ventricle is sensitive to increased RV pressure overload [36], RV function may not have been impaired before the administration of each pulmonary vasodilator.

The combination of BPS and sildenafil significantly decreased PVR and SVR to a greater extent than the application of either drug alone. These drugs cause vasodilating effects through different mechanisms: BPS selectively activates adenylate cyclase, elevates the cyclic adenosine monophosphate level in cells and blood platelets, and inhibits Ca^2+^ influx and thromboxane-A_2_ production [2,37,38,39], whereas sildenafil inhibits cyclic guanosine monophosphate [2,40]. Consequently, the differences in mechanism are thought to be synergistic, and emphasize the efficacy of multidrug therapy in cases with severe PH. Additionally, adding 15 µg/kg BPS to the 1.0 mg/kg sildenafil did not induce further left-heart enlargement or increase left-atrial pressure. Our results suggest that systemic vasodilation caused by BPS may prevent excessive left-heart load through increased pulmonary circulation in patients with PH secondary to left-heart disease. Further studies using more severe cases are required to establish the efficacy and safety of multidrug combination therapies with various pulmonary vasodilators.

This study had several limitations. First, we examined the hemodynamic and functional effects of single-drug doses of BPS and sildenafil. As both drugs are considered to have dose-dependent vasodilatory effects [15,41], further studies are needed to investigate their hemodynamic effects at higher doses and/or higher frequencies. Second, this study used canine models of MR induced by two methods with rupture of the chordae tendineae attaching to the anterior apex of the mitral valve. Although all dogs had moderate-to-severe MR and left-heart enlargement, the pathophysiology of MR might be different from the clinical case. Additionally, the pathophysiology of PH based on PAP and echocardiographic RV function was relatively mild. Our results might have changed if the dogs had more severe PH secondary to left-heart disease and RV dysfunction as seen in clinical practice. Furthermore, a small sample size might lower the statistical power to identify the differences. Finally, pulmonary vasodilators, especially BPS, have been reported to have differences in pharmacological efficacy among animal species [35]. Our results may not be fully applicable to species other than dogs.

## 5. Conclusions

Both 15 µg/kg BPS and 1.0 mg/kg sildenafil significantly decreased PAP and PVR, suggesting that these pulmonary vasodilators were effective for treating PH. However, differences in the pulmonary vasodilating effects of the two drugs were observed. Specifically, 1.0 mg/kg sildenafil had a more potent pulmonary vasodilating effect than 15 µg/kg BPS. In contrast, the systemic vasodilating effect of BPS was equivalent to its pulmonary vasodilating effect and stronger than that of sildenafil, which prevented excessive left-heart load. Additionally, combination therapy with BPS and sildenafil synergistically dilated pulmonary and systemic vessels. Furthermore, combination therapy prevented sildenafil-induced worsening of the left-heart loading condition, suggesting that it might be a more potent treatment option in severe PH cases.

## Figures and Tables

**Figure 1 biology-13-00311-f001:**
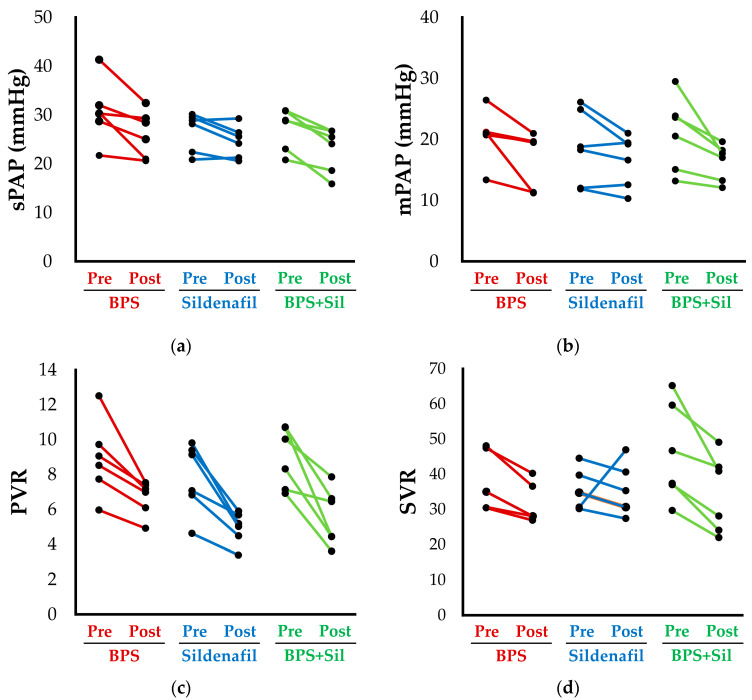
Individual plots before and after administrating pulmonary vasodilator: (**a**) systolic pulmonary arterial pressure (sPAP); (**b**) mean pulmonary arterial pressure (mPAP); (**c**) pulmonary vascular resistance (PVR); (**d**) systemic vascular resistance (SVR). Red, blue, and green lines represent the change in each value with pulmonary vasodilators (15 µg/kg beraprost sodium [BPS], 1.0 mg/kg sildenafil, and their combination [BPS + Sil], respectively).

**Figure 2 biology-13-00311-f002:**
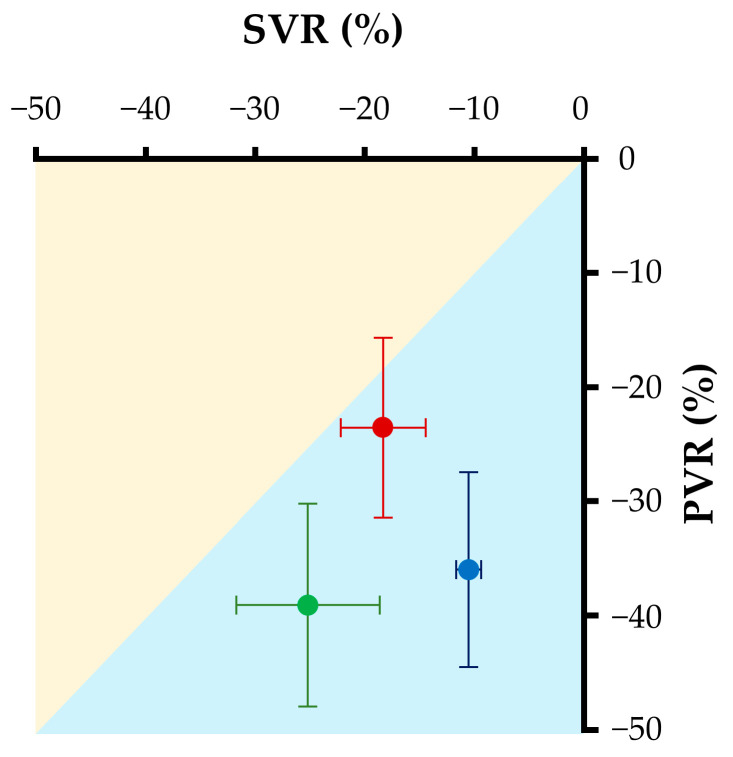
Scatter plots showing the decreasing rate of SVR and PVR. Red, blue, and green dots indicate the average values of decreasing rates after administrations of beraprost sodium, sildenafil, and BPS + Sil, respectively. Horizontal and vertical lines indicate the standard deviations. Orange area indicates systemic vasopredominant and blue area indicates pulmonary vasopredominant vasodilation effects. PVR, pulmonary vascular resistance; SVR, systemic vascular resistance.

**Table 1 biology-13-00311-t001:** Results of hemodynamical variables before and after administrating pulmonary vasodilators.

Variables	BPS (*n* = 6)		Sildenafil (*n* = 6)		BPS + Sil (*n* = 6)	
	Pre	Post	Pre	Post	Pre	Post
Heart rate (bpm)	86 (80, 89)	83 (75, 92)	86 (76, 87)	88 (81, 92)	72 (72, 85)	84 (74, 89)
SAP (mmHg)	120 (102, 125)	113 (104, 122)	114 (101, 117)	111 (99, 122)	115 (111, 126)	113 (107, 116)
MAP (mmHg)	82 (68, 96)	86 (73, 93)	79 (70, 87)	70 (61, 84)	84 (80, 92)	80 (74, 86)
PCWP (mmHg)	10.7 (8.8, 11.8)	10.2 (3.7, 11.2)	8.4 (5.8, 11.6)	11.3 (7.6, 12.7) *	9.4 (5.6, 12.6)	8.9 (5.2, 13.1)
sPAP (mmHg)	30.4 (27.0, 34.4)	26.8 (20.9, 30.1) *	28.5 (22.0, 29.7)	24.9 (21.1, 27.1) *	28.9 (22.5, 30.9)	24.8 (18.0, 26.7) *
mPAP (mmHg)	21.0 (18.9, 22.5)	19.6 (11.3, 20.0) *	18.6 (12.0, 25.2)	17.9 (12.0, 19.9) *	22.1 (14.6, 25.3)	17.4 (13.0, 18.6) *
dPAP (mmHg)	13.8 (11.6, 15.1)	12.8 (6.6, 13.0)	11.0 (6.4, 20.3)	8.2 (4.6, 12.4)	12.7 (9.3, 15.4)	11.0 (9.5, 11.5)
sRVP (mmHg)	28.8 (26.6, 37.0)	23.5 (20.2, 32.4) *	23.3 (21.5, 30.1)	21.8 (16.3, 28.8)	23.9 (18.4, 28.9)	22.9 (19.3, 27.3)
dRVP (mmHg)	1.0 (0.6, 3.0)	1.1 (0.4, 4.6)	2.3 (0.2, 3.8)	1.5 (0.3, 3.2)	2.1 (0.9, 4.0)	1.6 (0.8, 2.2)
dP/dt_max_ (mmHg/s)	377 (258, 490)	429 (350, 476)	350 (309, 408)	395 (342, 445) *	298 (256, 360)	379 (290, 424) *
dP/dt_min_ (mmHg/s)	−327 (−427, −229)	−382 (−396, −248)	−242 (−303, −169)	−317 (−356, −258) *	−261 (−317, −219)	−327 (−370, −261)
RAP (mmHg)	2.3 (0.7, 4.0)	3.3 (1.5, 5.7)	1.3 (0.7, 2.5)	1.7 (0, 2.9)	2.7 (1.9, 3.1)	2.7 (1.9, 3.5)
CVP (mmHg)	2.9 (0.7, 4.4)	2.1 (0.9, 3.6)	1.5 (0.4, 2.5)	2.3 (0.2, 3.7)	2.7 (1.3, 4.0)	1.5 (0.5, 4.4)
RV CO (L/min)	1.2 (1.0, 1.3)	1.3 (1.1, 1.5)	1.2 (1.1, 1.5)	1.4 (1.2, 1.5) *	1.1 (1.0, 1.6)	1.4 (1.1, 1.7) *
RV SV (mL)	14.0 (12.1, 15.6)	15.5 (14.1, 17.5) *	15.6 (12.8, 17.6)	15.9 (13.1, 17.3)	15.4 (12.4, 20.6)	17.8 (14.8, 19.5)
PVR	8.8 (7.3, 10.4)	7.1 (5.8, 7.4) *	8.1 (6.3, 9.5)	5.4 (4.2, 5.9) *	9.2 (7.8, 10.7)	5.4 (4.2, 6.9) *
SVR	35.1 (33.6, 47.6)	28.1 (27.8, 37.5) *	37.2 (33.6, 43.0)	33.0 (29.8, 38.7)	42.0 (35.2, 61.1)	34.5 (23.5, 43.8) *

Abbreviations: CVP, mean central venous pressure; dP/dt_max_, maximal first derivative of right-ventricular pressure; dP/dt_min_, minimal first derivative of right-ventricular pressure; dPAP, diastolic pulmonary arterial pressure; dRVP, diastolic right-ventricular pressure; MAP, mean systemic arterial pressure; mPAP, mean pulmonary arterial pressure; PCWP, pulmonary capillary wedge pressure; PVR, pulmonary vascular resistance; RAP, mean right-atrial pressure; RV CO, right-ventricular cardiac output; RV SV, right-ventricular stroke volume; SAP, systolic systemic arterial pressure; sPAP, systolic pulmonary arterial pressure; sRVP, systolic right-ventricular pressure; SVR, systemic arterial resistance. * This value is significantly different from before drug administration.

**Table 2 biology-13-00311-t002:** Results of echocardiographic variables for left heart before and after administrating pulmonary vasodilators.

Variables	BPS (*n* = 6)		Sildenafil (*n* = 6)		BPS + Sil (*n* = 6)	
	Pre	Post	Pre	Post	Pre	Post
LA/Ao	1.6 (1.5, 1.7)	1.6 (1.5, 1.8)	1.6 (1.6, 1.7)	1.6 (1.6, 1.7)	1.7 (1.5, 1.7)	1.6 (1.6, 1.7)
LVEDV (mL/m^2^)	42.1 (35.7, 52.4)	37.3 (33.3, 44.1)	41.6 (37.9, 48.2)	47.0 (42.1, 54.1) *	41.3 (35.7, 55.1)	39.3 (34.9, 52.3)
LVESV (mL/m^2^)	18.4 (15.5, 24.2)	15.5 (13.3, 18.2)	17.2 (13.8, 22.0)	18.7 (15.2, 25.5) *	18.5 (14.5, 29.3)	16.8 (10.8, 23.9) *
EF (%)	51.4 (45.6, 64.6)	61.3 (50.1, 68.4)	60.5 (50.4, 65.4)	62.1 (49.5, 65.4)	54.2 (35.3, 66.0)	61.1 (42.7, 71.0) *
E (m/s)	0.9 (0.8, 1.1)	1.0 (0.9, 1.1)	1.0 (0.8, 1.3)	1.0 (0.9, 1.4)	0.9 (0.8, 1.1)	0.9 (0.9, 1.2)
A (m/s)	0.3 (0.3, 0.4)	0.4 (0.3, 0.5)	0.4 (0.3, 0.4)	0.3 (0.3, 0.4)	0.4 (0.3, 0.4)	0.3 (0.3, 0.4)
E/A	2.9 (2.3, 3.5)	2.3 (2.1, 3.5)	2.9 (2.3, 3.0)	3.1 (2.5, 4.2)	2.9 (2.1, 3.6)	2.7 (2.2, 4.5)
E/e’	10.8 (9.2, 12.0)	10.2 (8.5, 11.3)	11.7 (11.0, 12.3)	13.5 (10.0, 16.5) *	10.3 (8.4, 11.3)	11.1 (9.0, 13.2)
LV CO (mL)	2.0 (1.8, 2.3)	2.5 (2.0, 3.0) *	2.0 (1.6, 2.2)	2.3 (2.1, 2.5)	1.9 (1.5, 2.3)	2.4 (1.7, 3.7) *
LV-SL (%)	18.5 (16.5, 19.6)	20.1 (18.6, 23.4)	18.7 (17.0, 20.3)	20.7 (18.0, 22.0) *	17.5 (15.7, 18.7)	20.4 (17.6, 22.8)
LV-SC (%)	19.1 (17.6, 21.5)	18.8 (17.7, 22.2)	18.0 (17.5, 22.0)	18.9 (17.2, 25.4)	21.0 (18.7, 23.9)	20.9 (16.6, 27.0)

Abbreviations: A, late-diastolic transmitral flow velocity; E, early-diastolic transmitral flow velocity; e’, early-diastolic myocardial velocity of the septal mitral annulus; EF, ejection fraction; LA/Ao, left-atrial-to-aortic diameter ratio; LV CO, left-ventricular cardiac output; LVEDV, end-diastolic left-ventricular volume; LVESV, end-systolic left-ventricular volume; LV-SC, left-ventricular circumferential strain; LV-SL, left-ventricular longitudinal strain. * This value is significantly different from before drug administration.

**Table 3 biology-13-00311-t003:** Results of echocardiographic variables for right heart before and after administrating pulmonary vasodilators.

Variables	BPS (*n* = 6)		Sildenafil (*n* = 6)		BPS + Sil (*n* = 6)	
	Pre	Post	Pre	Post	Pre	Post
RVEDA (cm^2^)	4.9 (4.4, 5.2)	4.6 (4.1, 5.1)	4.8 (4.3, 5.3)	4.4 (4.2, 5.0)	4.6 (4.2, 5.0)	4.6 (4.5, 5.2)
RVESA (cm^2^)	2.7 (2.4, 3.0)	2.5 (2.1, 2.7)	2.8 (2.5, 3.1)	2.4 (2.2, 2.7) *	2.4 (2, 2.9)	2.5 (2.2, 3.4)
RV FACn (%/kg^−0.097^)	57.1 (51.4, 64.7)	60.2 (50.0, 67.9)	52.0 (46.9, 60.3)	59.4 (54.3, 65.3) *	61 (52.4, 64.9)	61.0 (56.3, 66.0)
TAPSEn (mm/kg^0.284^)	5.5 (5.3, 5.7)	6.4 (5.6, 6.7)	5.8 (5.1, 6.2)	6.1 (5.9, 6.8)	5.8 (5.1, 6.4)	6.5 (5.1, 7.3) *
RV s’ (cm/s)	9.5 (8.6, 10.3)	9.2 (8.2, 10.0)	9.6 (8.9, 10.1)	10.1 (9.7, 10.7)	9.6 (8.3, 10.3)	9.1 (8.5, 10.6)
RV-SL_3seg_ (%)	21.1 (18.6, 27.1)	22.9 (18.6, 24.9)	22.1 (20.9, 25.5)	24.1 (23.1, 26.4)	21.3 (19.5, 24.0)	25.7 (20.9, 27.8) *
RV-SL_6seg_ (%)	19.3 (16.6, 21.6)	20.8 (17.5, 21.9)	18.7 (17.7, 21.3)	20.6 (19.1, 21.6)	20.2 (16.8, 22.3)	21.7 (20.0, 25.5) *

Abbreviations: 3seg, only right-ventricular free-wall analysis; 6seg, entire right-ventricular analysis; RV FAC, right-ventricular fractional area change; RV s’, systolic myocardial velocity of the lateral tricuspid annulus; RVEDA, end-diastolic right-ventricular area; RVESA, end-systolic right-ventricular area; RV-SL, right-ventricular longitudinal strain; TAPSE, tricuspid annular plane systolic excursion. * This value is significantly different from before drug administration.

## Data Availability

The datasets used or analyzed in the current study are available from the corresponding author upon reasonable request.

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
