# Peer review of "Comparative Study of Cardiovascular Effects of Selected Pulmonary Vasodilators in Canine Models of Mitral Valve Disease"

_biology, 2024, doi:10.3390/biology13050311_

Round 1

Reviewer 1 Report

Comments and Suggestions for Authors

This is an animal study, which compared the hemodynamic and functional effects of selected pulmonary vasodilators (beraprost sodium (BPS) and sildenafil), and evaluated the clinical efficacy of combination therapy with BPS and sildenafil in dogs with experimentally induced mitral regurgitation (MR). The authors demonstrated that both 15 μg/kg BPS and 1.0 mg/kg sildenafil significantly decreased pulmonary artery pressure and pulmonary vascular resistance, suggesting that these pulmonary vasodilators were effective for treating pulmonary hypertension (PH), and that the differences in the pulmonary vasodilating effects between the two drugs were observed. Sildenafil of 1.0 mg/kg had more potent pulmonary vasodilating effect than 15 μg/kg BPS. In contrast, the systemic vasodilating effect of BPS was equivalent to its pulmonary vasodilating effect and stronger than that of sildenafil, which prevented excessive left-heart load. Additionally, combination therapy with BPS and sildenafil synergistically dilated pulmonary and systemic vessels. Furthermore, combination therapy prevented sildenafil-induced worsening of the left-heart-loading condition. 

The reviewer considers that the authors have well performed this experiment, despite the established distinctions between BPS and sildenafil in clinical practice. This reviewer has a significant concern as outlined below:

Major comments:

1.       At the end of conclusion section, the authors mentioned that their results would be important to decide the treatment plan using pulmonary vasodilator especially in patients with PH secondary to left heart disease. While the use of sildenafil, albeit off-label, may be considered acceptable, it is well-documented that BPS is ineffective in treating PH associated with left heart disease. Generally, prostacyclin is contraindicated in cases of PH due to left heart conditions. The authors should remove this issue throughout the manuscript. 

Author Response

Response to Reviewer 1

Dear Reviewer 1

Comments and Suggestions for Authors

This is an animal study, which compared the hemodynamic and functional effects of selected pulmonary vasodilators (beraprost sodium (BPS) and sildenafil), and evaluated the clinical efficacy of combination therapy with BPS and sildenafil in dogs with experimentally induced mitral regurgitation (MR). The authors demonstrated that both 15 μg/kg BPS and 1.0 mg/kg sildenafil significantly decreased pulmonary artery pressure and pulmonary vascular resistance, suggesting that these pulmonary vasodilators were effective for treating pulmonary hypertension (PH), and that the differences in the pulmonary vasodilating effects between the two drugs were observed. Sildenafil of 1.0 mg/kg had more potent pulmonary vasodilating effect than 15 μg/kg BPS. In contrast, the systemic vasodilating effect of BPS was equivalent to its pulmonary vasodilating effect and stronger than that of sildenafil, which prevented excessive left-heart load. Additionally, combination therapy with BPS and sildenafil synergistically dilated pulmonary and systemic vessels. Furthermore, combination therapy prevented sildenafil-induced worsening of the left-heart-loading condition.

The reviewer considers that the authors have well performed this experiment, despite the established distinctions between BPS and sildenafil in clinical practice. This reviewer has a significant concern as outlined below:

Response: We wish to express our strong appreciation to the Reviewer for their insightful comments on our paper. We feel the comments have helped us significantly improve the paper. We hope that the revised paper meets your approval and will be more suitable for publication in biology for the Article. Please see the following point-by-point responses for details.

Major comments:

  1. At the end of conclusion section, the authors mentioned that their results would be important to decide the treatment plan using pulmonary vasodilator especially in patients with PH secondary to left heart disease. While the use of sildenafil, albeit off-label, may be considered acceptable, it is well-documented that BPS is ineffective in treating PH associated with left heart disease. Generally, prostacyclin is contraindicated in cases of PH due to left heart conditions. The authors should remove this issue throughout the manuscript.

Response: Thank you very much for your comment. As you mentioned, it is documented that pulmonary vasodilator, including beraprost, is ineffective in treating human PH. However, as is reported in the following report, it is also reported that the species differences influence the pharmacologic effects of beraprost.

Nishio, S.; Matsuura, H.; Kanai, N.; Fukatsu, Y.; Hirano, T.; Nishikawa, N.; Kameoka, K.; Umetsu, T. The in Vitro and Ex Vivo Antiplatelet Effect of TRK-100, a Stable Prostacyclin Analog, in Several Species. Jpn. J. Pharmacol. 1988, 47, 1–10, doi:10.1254/jjp.47.1.

Therefore, we consider that beraprost would be effective especially in dogs with PH. We have revised our discussion and conclusions because this opinion might only be applicable to dogs.

Line 14-15: Pulmonary hypertension is a fatal comorbidity in dogs with left-sided heart disease.

Line 25-26: This study demonstrated the differences between beraprost and sildenafil in pulmonary and systemic vasodilating effects.

Line 42-44: Both beraprost and sildenafil were effective against canine PH; however, sildenafil was associated with the risk of worsening left-heart loading.

Line 59-61: However, pulmonary vasodilators are typically considered in cases of PH that cannot be controlled solely by treating underlying diseases in dogs.

Line 298-303: These results indicated that BPS could potentially treat canine PH caused by left-heart disease more safely than sildenafil. However, BPS has reported animal species differences in the pharmacologic effects. Our results might be different in the other species than dogs, as reported in PH guideline that BPS showed no hemodynamic improvements or long-term outcome benefit in human patients with PH.

Line 337-339: Finally, pulmonary vasodilators, especially BPS, have been reported to have differences in pharmacological efficacy among animal species. Our results may not be fully applicable to species other than dogs.

Reviewer 2 Report

Comments and Suggestions for Authors

Thank you for providing me the opportunity to review the paper entitled " Comparative study of cardiovascular effect of selected pulmonary vasodilators in canine models of mitral valve disease".

General comments:

This study compared the hemodynamic and functional effects of selected pulmonary vasodilators (BPS and sildenafil) and evaluated the clinical efficacy of combination therapy with BPS and sildenafil in dogs with experimentally induced mitral regurgitation (MR). The authors hypothesized that the degree of pulmonary vasodilating effects of BPS and sildenafil were different, and that sildenafil could improve PH pathophysiology but worsen left-heart-loading conditions. Furthermore, combination therapy with BPS and sildenafil could be a more potent treatment for PH, owing to their synergistic effects. And they concluded that Both 15 µg/kg BPS and 1.0 mg/kg sildenafil significantly decreased PAP and PVR, suggesting that these pulmonary vasodilators were effective for treating PH. However, differences in the pulmonary vasodilating effects of the two drugs were observed. Specifically, 1.0 mg/kg sildenafil had more potent pulmonary vasodilating effect than 15 µg/kg BPS. In contrast, the systemic vasodilating effect of BPS was equivalent to its pulmonary vasodilating effect and stronger than that of sildenafil, which prevented excessive left-heart load. It is a well-written manuscript, and the findings are well presented. I have some comments.

Minor:

1. in this study, the authors used total 6 dogs. Is there any difference of baseline characteristics such as body weight among dogs. And please add on those

2. In Table 1, 2, the authors showed results of hemodynamical & echo variables according to drugs combinations. I just wonder if these 3 groups are divided such as 2 dogs for BPS; 2 dogs for Sildenafil; 2 dogd for BPS+Sil. Please explain

Comments on the Quality of English Language

minor revision can be needed.

Author Response

Response to Reviewer 2

Dear Reviewer 2

Comments and Suggestions for Authors

General comments:

This study compared the hemodynamic and functional effects of selected pulmonary vasodilators (BPS and sildenafil) and evaluated the clinical efficacy of combination therapy with BPS and sildenafil in dogs with experimentally induced mitral regurgitation (MR). The authors hypothesized that the degree of pulmonary vasodilating effects of BPS and sildenafil were different, and that sildenafil could improve PH pathophysiology but worsen left-heart-loading conditions. Furthermore, combination therapy with BPS and sildenafil could be a more potent treatment for PH, owing to their synergistic effects. And they concluded that Both 15 µg/kg BPS and 1.0 mg/kg sildenafil significantly decreased PAP and PVR, suggesting that these pulmonary vasodilators were effective for treating PH. However, differences in the pulmonary vasodilating effects of the two drugs were observed. Specifically, 1.0 mg/kg sildenafil had more potent pulmonary vasodilating effect than 15 µg/kg BPS. In contrast, the systemic vasodilating effect of BPS was equivalent to its pulmonary vasodilating effect and stronger than that of sildenafil, which prevented excessive left-heart load. It is a well-written manuscript, and the findings are well presented. I have some comments.

Response: We wish to express our strong appreciation to the Reviewer for their insightful comments on our paper. We feel the comments have helped us significantly improve the paper. We hope that the revised paper meets your approval and will be more suitable for publication in biology for the Article. Please see the following point-by-point responses for details.

Minor:

  1. in this study, the authors used total 6 dogs. Is there any difference of baseline characteristics such as body weight among dogs. And please add on those

Response: Thank you very much for your comment. As noted in “2.1. Animals and Study preparation” and “3. Results”, there were no substantial differences in clinical characteristics, such as age (2.4 ± 0.1 years; range: 2.2–2.7), body weight (11.9 ± 0.8 kg; range: 10.5–13.3), and clinical stage of mitral valve disease (all dogs were ACVIM Stage B2). Additionally, because the same 6 dogs were used in all groups and were studied by a crossover method, we consider that the differences among groups were minimal. We have clarified the information according to the reviewer’s comment.

Line 117-118: Each drug was tested using all 6 dogs by a crossover method.

Line 199-203: At the beginning of the study, all canine models had moderate to severe MR and left-heart enlargement without any evidence of heart failure, indicating American College of Veterinary Internal Medicine Stage B2 (mean ± standard deviation of LA/Ao and the normalized end-diastolic LV internal dimensions at the beginning of study were 1.7 ± 0.1 and 1.8 ± 0.1, respectively).

  1. In Table 1, 2, the authors showed results of hemodynamical & echo variables according to drugs combinations. I just wonder if these 3 groups are divided such as 2 dogs for BPS; 2 dogs for Sildenafil; 2 dogd for BPS+Sil. Please explain

Response: Thank you very much for your comment. We have examined the hemodynamic effects of beraprost, sildenafil, and their combination by a crossover method using all 6 dogs. Therefore, each group consisted of 6 dogs. We have clarified the methods and tables according to the reviewer’s comment.

Line 117-118: Each drug was tested using all 6 dogs by a crossover method.

Round 2

Reviewer 1 Report

Comments and Suggestions for Authors

This reviewer has no further comment.